# First-line infertility treatment in normal or subnormal sperm: Interest of a simplified pre-IMSI test

Julien Sigala[1]*, Sophie Poirey[1], Julien Robert[2], Olivier Pouget[3], Thibault Mura[2], Stephanie Huberlant[3,4], Nathalie Rougier[1]

1 Department of Reproductive Biology, Nîmes University Hospital, Nîmes, France, 2 Department of Biostatistics, Epidemiology, Public Health and Innovation in Methodology, Nîmes University Hospital, Nîmes, France, 3 Department of Obstetrics and Gynecology, Nîmes University Hospital, Nîmes, France, 4 University of Montpellier-Nîmes, Nîmes, France

* julien.sigala@chu-nimes.fr

## Abstract

### Background

In the field of male infertility, when sperm is normal/subnormal, a few "add-on" routine tests can complete the basic semen examination.

### Objectives

The aim of this study was to develop and evaluate a faster, simplified motile sperm organelle morphology examination (MSOME) technique for selected infertile patients with apparently normal/subnormal sperm and, in their background: failure of two or three intrauterine insemination (IUI) cycles, repeatedly fragmented embryos, embryonic development to blastocyst-stage failures, repeated miscarriages, a long period of infertility or 2 or more IVF attempts without pregnancy. Our test results were correlated with IUI, conventional *in vitro* fertilization (IVF), intracytoplasmic sperm injection (ICSI) and intracytoplasmic morphologically selected sperm injection (IMSI) outcomes.

### Materials and methods

We validated an adapted version of the MSOME analysis called the pre-IMSI test (PIT), based on vacuole evaluation alone. 248 infertile patients from our assisted reproductive technology (ART) Center were retrospectively selected and split into three PIT score subgroups (patients with ≤8% (score I), 9 to 15% (score II) and ≥16% normal spermatozoa (score III)) based on the correlation between PIT results and each ART technique outcome. The choice of one or another of these ART techniques had been made according to the usual clinico-biological criteria.

### Results

Clinical outcomes for each of the three PIT subgroups were compared individually for the different ART techniques. For ICSI, the effect of the PIT score subgroup was significant for

**Data Availability Statement:** All relevant data are within the manuscript and its Supporting Information files.

**Funding:** The author(s) received no specific funding for this work.

**Competing interests:** The authors have declared that no competing interests exist.

clinical pregnancies (p = 0.0054) and presented a trend for live births (p = 0.0614). Miscarriage rates of IVF attempts were statistically different depending on the PIT score (p = 0.0348). Furthermore, the odds ratios of clinical pregnancy rates were significantly different according to PIT score subgroup when comparing ICSI *vs.* IMSI or IVF *vs.* ICSI attempts.

## Discussion

IMSI appears to be recommended when sperm belongs to PIT score I, ICSI when it belongs to PIT score II and IVF or IUI when sperm is of PIT score III quality in selected infertile couples. The lack of statistical power in these PIT subgroups means that we must remain cautious in interpreting results.

## Conclusion

Our results support the interest of this simplified test for certain couples with normal/subnormal sperm to help choose the most efficient ART technique, even as first-line treatment.

## Introduction

In the field of male fertility exploration, a few "add-on" routine tests can be made to complete the basic semen examination [1], like high magnification motile sperm organelle morphology (MSOME) [2], sperm deoxyribonucleic acid (DNA) fragmentation [3] and sperm DNA base oxidation evaluations [4]. Sperm DNA integrity evaluations can only be used at diagnosis, not in real time during ART attempts. Evaluating spermatozoa at high magnification remains a good non-invasive strategy to complete the initial examinations even for couples with apparently normal/subnormal sperm. It could help embryologists choose the most suitable ART technique to use, particularly IMSI when this is required.

MSOME is not a recent technique but, despite a significant number of studies on the subject, there is a lack of consensus on results and the ensuing decisions [2, 5–10]. The origin of nuclear vacuoles in male infertility remains also widely debated [11]. Early reports suggest that sperm head vacuoles may be due to DNA fragmentation [12–17], DNA decondensation [18–20] or exposure to reactive oxygen species (ROS) [21, 22]. Previous results [12, 13, 15–17, 20, 23, 24] show that the DNA fragmentation rate may reflect the size and number of vacuoles, though the role of vacuoles in the process of DNA fragmentation is not yet fully understood. Moreover, spermatozoa with vacuoles have been shown to be associated with chromatin disorganization (abnormal chromatin condensation, aneuploidy, modification of spatial and chromosomal positioning) [2, 13, 18, 25, 26]. A potential association between spermatozoa genomic stability and vacuolar morphology and location was also published [27] as well as the potential link between vacuolated spermatozoa and some epigenetic marks involved in chromatin condensation [28, 29]. Conversely, some studies argue that nuclear vacuoles result from a natural physiological process unrelated to DNA fragmentation [30, 31]. In all cases, the use of high-magnification sperm selection to discard vacuolated sperm prior to microinjection might improve sperm selection and the outcomes of assisted reproduction techniques [32–34].

In practice, even today the indications for MSOME or IMSI are not clearly identified and these techniques are still considered as too time-consuming. Besides, the cut-off for classifying spermatozoa as "usable" or "non-usable" remains unclear and the classifications for normal or abnormal MSOME spermatozoa vary. Thus, assessing the size, number and location of

vacuoles differs greatly from one study to another [17]. Recently, it has been suggested that the depth and location of vacuoles should also be included in the evaluation criteria [27]. The lack of consensus on the various classifications is also clear in the decision-making about the indications for MSOME and IMSI. In a recent review, Mangoli *et al.* [35] concluded that only patients with repeated implantation failures, severe male infertility factors or advanced male or maternal ages had higher chances of conceiving with IMSI. However, only a few articles in the literature focus on the use of MSOME and IMSI on normal/subnormal sperm [36–40].

In the light of all this, our study aimed to (1) develop and (2) evaluate a faster, simplified MSOME technique for routine use on the day of oocyte retrieval. We then (3) retrospectively confronted our test results with each ART technique outcome i.e. IUI, IVF, ICSI and IMSI, in patients with apparently normal/subnormal sperm.

## Materials and methods

### Patients

A total of 248 infertile couples from the IVF Center at Nîmes University Hospital, France, were finally included in the present study from February 2017 to October 2020.

### Ethics statement

All patients eligible for inclusion in the study had received a letter of information and non-opposition to the use of their health data. The procedure was approved by the local institutional review board at Nîmes University Hospital (IRB n° 21.12.01).

Only two authors (first and last) had access to information that could identify individual participants during and after data collection. These authors were also the embryologists of the patients included in this study.

The data consulted for research purposes in this study was done throughout the study period, from the start-up of the database (November 2020) to the production of the final statistics (October the 26th 2023).

### Study design

We opted for an evolving methodology. An adapted version of the MSOME analysis called the pre-IMSI test (PIT), first based on well-known publications [5, 39], was systematically proposed to all eligible patients (n = 402) during the study period when sperm was normal or subnormal according to the WHO laboratory manual (2010) [41]. Subnormal sperm may present one or more minor abnormalities in the spermogram-spermocytogram, but at least 1 million progressive spermatozoa were available after selection. All sperms were of sufficient quality to technically perform IUI, IVF, ICSI or IMSI, and all couples were eligible for each of these techniques. Indications for PIT were: having normal/subnormal sperm associated with unexplained infertility, failure of two or three IUI cycles, repeatedly fragmented embryos, embryonic development culture failures at blastocyst-stage, repeated miscarriages, a long period of infertility and two or more IVF attempts without pregnancy.

We first validated the methodology to develop a simplified, efficient, reproducible technique based on 402 eligible patients. We then excluded 128 patients who had not yet done ART attempts, 26 women aged over 39 years old or patients with cryopreserved semen. Women aged 39 or over were excluded to avoid an oocyte quality bias and because, in our experience, IMSI does not offer better pregnancy outcomes (unpublished data). Only 248 couples were finally included in our study to compare PIT results and ART outcomes. We were able to individualize three PIT score patient subgroups (patients with ≤8% normal

spermatozoa (score I), those with a 9 to 15% normal spermatozoa (score II) and those with a ≥16% normal spermatozoa (score III)). The choice of one or another of these ART techniques had only been made according to the usual clinico-biological criteria.

### Semen samples and quality assessment

Sperm samples were collected by masturbation after 2 to 7 days of sexual abstinence. Samples were analyzed by microscopy at x40 magnification after half to one hour of liquefaction. Sperm concentration (c-chip disposable hematocymeter [Malassez]), motility and morphology was assessed according to the WHO 2010 criteria [41]. Semen characteristics are summarized in Table 1. Since 2019, all results of our ART laboratory are delivered under accreditation according to the ISO 15189 international standard. All sperms were prepared by the discontinuous density gradient technique. After liquefaction at room temperature, the semen was overlaid on a 40% and 80% SupraSperm® (CooperSurgical, Denmark) gradient column and centrifuged at 350g for 20 minutes. The sperm pellet was washed at 600g in universal IVF medium (CooperSurgical, Denmark). The seminal fluid, the 40% phase, and the upper half of the 80% phase were gently aspirated. Then, 3 mL of universal IVF medium (CooperSurgical, Denmark) was gently layered over the lower half of the 80% phase and the sample was centrifuged 10 min at 600g. The supernatant was removed to obtain 0.2 to 0.4 ml of the preparation. Selected sperm preparation was store at 37˚C.

### Pre-IMSI test (PIT)

For the PIT, a SpermSlow™ drop (CooperSurgical, Denmark) was placed in a glass-bottomed Fluorodish (WPI, USA). The drop was covered with sterile mineral oil (CooperSurgical, Denmark). The selected sperm preparation was then placed in the SpermSlow™ drop. Sperm cells

**Table 1.  Main characteristics of the study population according to the three PIT score subgroups.**

|  | PIT score I | PIT score II | PIT score III | p |
|---|---|---|---|---|
| Couples (n) | 40 | 59 | 149 |  |
| Woman's age (years) | 33 [31;35.5] | 32 [29;36] | 32.5 [29;35] | 0.3767 |
| BMI | 22.5 [20;25] | 23 [21;27] | 23 [20;26] | 0.7545 |
| AMH (ng/L) | 2.1 [1.6;4] | 3.0 [1.7;4.2] | 2.7 [1.4;5] | 0.5360 |
| Duration of infertility (years) | 3 [2.5;5] | 3 [2;4] | 3 [2;4] | 0.4114 |
| Cause of infertility (%) |  |  |  |  |
| *Unexplained* | 10 | 23.7 | 17.4 | 0.2122 |
| *Tubal factor* | 20 | 8.5 | 13.4 | 0.2529 |
| *Ovulatory* | 5 [a] | 20.3 [a,b] | 24.8 [b] | 0.0224 |
| *Endometriosis* | 5 | 5.1 | 14.1 | 0.0846 |
| *Male factor* | 80 [a] | 55.9 [a,b] | 42.3 [b] | <0.0001 |
| *Male factor (except teratozoospermia)* | 27.5 | 27.1 | 25.5 | 0.9521 |
| Sperm parameters WHO 2010 |  |  |  |  |
| *Sperm concentration ($10^6$/mL)* | 23.7 [a] [11;51] | 29 [a,b] [11;79] | 44 [b] [20;73] | 0.0111 |
| *Total sperm count ($10^6$)* | 77.4 [a] [33;157] | 109.7 [a,b] [41;220] | 158.6 [b] [74;264] | 0.0027 |
| *Progressive motility (%)* | 48 [39;56] | 48 [34;58] | 49 [36.5;60] | 0.8985 |
| *Normal morphology (%)* | 1.5 [1;3] | 4 [1;5] | 6 [3.5;10] | <0.0001 |

Values are expressed as median and quartiles [q1;q3]. *p<0.05* is considered significant.

AMH = anti-Müllerian hormone; BMI = body mass index; PIT = pre-IMSI test; WHO = World Health Organization

were observed under an inverted microscope (Leica DMI8) equipped with Nomarski differential interference contrast optics. The images were captured by a TK-C1481BEG color video camera (JVC professional). Magnification ranged from x6600 to x12000.

The sperm cell grading system used was adapted from the Vanderzwalmen classification [5]. It was based on the number and size of vacuoles (Fig 1), without considering sperm morphology. Like Vanderzwalmen *et al.*, we first individualized four sperm grades and then, due to unreliable inter-operator CVs, we split them into two (grade A and grade B). Grade A sperm cells either presented no vacuole, a maximum of two small vacuoles or an intermediate vacuole. Grade A were considered as normal spermatozoa. Grade B sperm cells had either more than two small vacuoles or at least one large vacuole. Grade B were considered as abnormal spermatozoa. Evaluation of vacuole size (small, intermediate, large) was semi-quantitative, as described in Fig 1.

The PIT was done by two operators who each analyzed a total of two counts of one hundred motile sperm at high magnification. The PIT score corresponded to the average grade A sperm count of the two operators. In the event of unreliable inter-operator CVs, a third operator also performed two counts of one hundred motile spermatozoa. This technique was validated in 2019 by the French accreditation committee (COFRAC) according to ISO 15189 international standards.

Finally, our PIT score interpretation was adapted from the Wittemer system [42], according to the percentage of grade A sperm cells. We first individualized four PIT scores; when we compared our PIT results and each ART technique outcome, there was no difference between the 2 intermediate groups. As a consequence, we decided to individualize three PIT scores: PIT score I corresponding to $\leq 8\%$ of grade A sperm cells, PIT score II corresponding to 9 to 15% of grade A sperm cells and PIT score III corresponding to $\geq 16\%$ of grade A sperm cells.

The couples included were then separated into three subgroups according to their PIT score. At this stage, PIT score I was considered as the worst quality, PIT score II as of intermediate quality and PIT score III as the best.

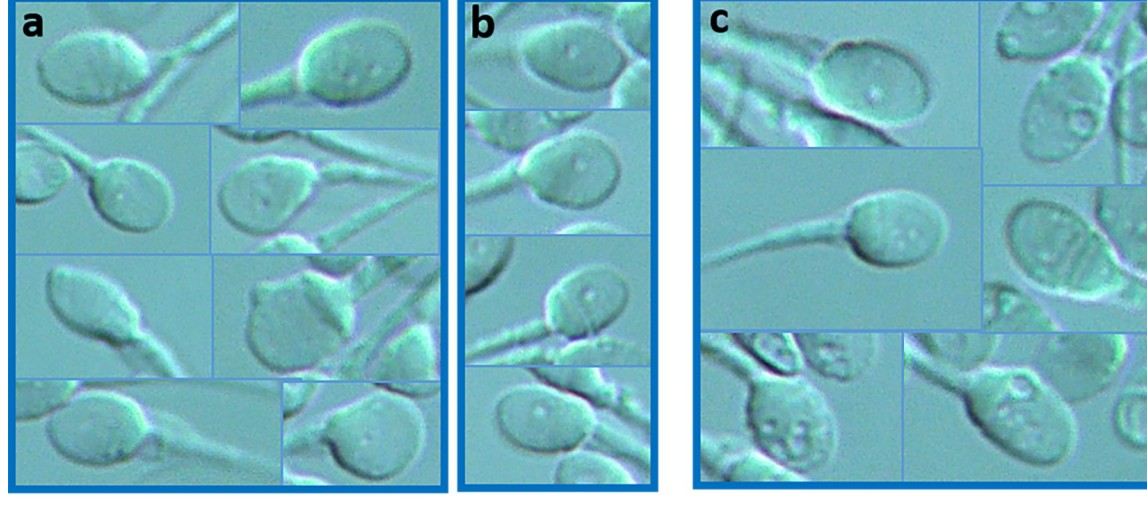

**Fig 1. Pre-IMSI test (PIT): grades of spermatozoa according to the number and size of vacuoles.** Grade A sperm cells either presented a maximum of two small vacuoles (a) or an intermediate vacuole (b). Grade B sperm cells had either more than two small vacuoles or at least one large vacuole (c).

### Assisted reproductive technology procedure

**IUI attempts.** One to 20 millions prepared progressive spermatozoa were inseminated 36 hours after hCG administration.

**IVF, ICSI and IMSI attempts.** In IVF attempts, oocytes were inseminated approximately 2–3 hours after oocyte retrieval. During ICSI/IMSI attempts, selected spermatozoa were injected under x400 magnification. For ICSI, motile and morphologically normal spermatozoa were injected. For IMSI, spermatozoa were chosen according to sperm selection method of Berkovitz *et al.* [27]. Briefly, normal morphological spermatozoa with no vacuoles or a maximum of two small vacuoles were used. Spermatozoa that presented deep vacuoles and/or vacuoles either at the nucleus or equatorial segment were systematically discarded. Injected oocytes and embryos were cultured either in Global® medium (JCD Laboratoires, France) or Cleav™/Blast™ medium (CooperSurgical, Denmark) in tri-gas incubators. Assessment of normal fertilization was estimated 17 ±1 hours post-insemination (IVF) or post-injection (ICSI/IMSI) [43]. Early cleavage rate was checked 25 hours post-insemination or post-injection. Cleaved embryo quality was estimated 44 to 46 hours post-fertilization. The embryo score (Blefco classification [44]) was established on Day 2 (44 ± 1 hours post- fertilization) or at the 8-cell stage on Day 3 (68 ± 1 hours post- fertilization). Blastocysts were evaluated on Days 5 and 6 according to the Gardner classification [45].

**Embryo transfer.** Embryo transfers were also assisted by ultrasound control with Elios (Ellios Cat Tek, France), Set TDT® (CCD laboratory, France) or Echogyn Embryoview M-18cm (CCD laboratory, France) catheters from Day 2 to Day 5 post-oocyte retrieval. The number of embryos transferred was decided according to the consensus of the American Society for Reproductive Medicine [46]. Clinical pregnancy was defined as an ongoing pregnancy confirmed by the number of gestational sacs with embryo and heart activity using ultrasound. Early miscarriages or pregnancy arrest were defined as the termination of a pregnancy during the first trimester.

### Statistical analysis

Statistical analysis was performed at Nîmes University Hospital using Statistical Analysis System software (SAS Institute, Cary, NC, USA) version 9.4 without provision for replacing missing data. A 5% alpha level of significance was used for all tests.

The initial data analysis was a description of the total population and per group. Statistical results were presented as means and standard deviations for quantitative variables with a Gaussian distribution, and medians and quartiles [Q1; Q3] for the other variables. For qualitative variables, the numbers and associated percentages were presented. The subgroup categories were compared according to PIT results. When the variables were quantitative, an Anova test was applied if the application conditions permitted it. A Kruskal-Wallis test was used where appropriate. For qualitative variables, a Chi-squared test was used when the application conditions permitted, otherwise a generalized Fisher's exact test (Fisher-Freeman-Halton exact) was used.

The clinical outcomes (ongoing pregnancy and live birth) of three PIT score subgroups for each of four different ART techniques were compared using mixed logistics models with the couple as a random effect. Miscarriages were expressed per ongoing pregnancy and compared using the Chi-squared test or Fisher's exact test as appropriate. Another mixed logistics model with the couple as a random effect including PIT score subgroups and ART techniques was used to compare rates of ongoing pregnancies between ART techniques in PIT score subgroups. In addition, results of interaction between PIT score subgroups and ART techniques were used to estimate the odds ratio (OR) of two-by-two ART technique comparisons in each PIT score subgroup. These ORs were then compared between PIT score subgroups.

## Results

The main characteristics of the patients under study are shown in Table 1. Sperm concentrations and total sperm counts differed significantly between PIT score I and PIT score III subgroups even though these parameters were considered as normal according to the WHO 2010 laboratory manual [41]. We observed that the morphology was different between PIT score I and II subgroups, between PIT score I and III subgroups and between PIT score II and III subgroups. In PIT score I, the morphology score (1.5%) was considered as abnormal (normal if >4% of typical spermatozoa). Among the causes of infertility, we noted a significant difference in ovulatory factors between the three subgroups.

Cycle characteristics are presented in Table 2. The number of progressive spermatozoa inseminated was similar in each subgroup. We noted a significant difference in the number of ICSI cycles between PIT score II and III subgroups. No IMSI was proposed when the sperm was of PIT score III. Analysis of the number of embryos obtained and transferred showed that these were equivalent in each subgroup. Embryos were transferred between Day 2 and Day 5 according to our Center's policy.

Clinical outcomes according to the three PIT subgroups were compared within the different ART techniques (Tables 3 and 4). The three PIT subgroups were compared with each other, using clinical pregnancy, miscarriage and live birth rates as reference points. For IUI, IVF and IMSI techniques, the PIT score subgroup appeared to have no effect on clinical outcomes (ongoing pregnancy and live birth). For ICSI, the effect of PIT score subgroup was significant for clinical pregnancies ($p = 0.0054$) and presented a statistical trend for live births

**Table 2. Cycle characteristics of the three PIT score subgroups.**

|  | PIT score I | PIT score II | PIT score III | *p* |
|---|---|---|---|---|
| **IUI** <br> **cycles *n* (%)** | **15 (12.2)** | **32 (26)** | **76 (61.8)** | |
| number of motile spermatozoa inseminated (millions) | 10.9 [9;14] | 10 [6;14] | 11.2 [8;13] | 0.9699 |
| **IVF** <br> **cycles *n* (%)** | **7 (5.4)** | **23 (17.8)** | **99 (76.8)** | |
| retrieved oocytes | 9.5 [6;12] | 11.5 [8;14] | 11 [8;16] | 0.5536 |
| MII oocytes | 8 [5;11] | 10 [7;12] | 9 [5;13] | 0.6879 |
| available embryos | 3 [2;4] | 5.7 [3;9] | 6 [3;10] | 0.2314 |
| embryo transferred | 1.4 [0;2] | 1 [1;1.6] | 1 [1;1.4] | 0.5787 |
| **ICSI** <br> **cycles *n* (%)** | **32 (17.9)** | **52 (29.0)** | **95 (53.1)** | |
| retrieved oocytes | 10.3 [7;13] | 11 [8;14] | 10 [7;13] | 0.5376 |
| MII oocytes | 8 [5;10] | 9 [6;12] | 7.3 [5;10] | 0.3102 |
| available embryos | 5 [2;6] | 5.5 [3;8] | 4.6 [3;6] | 0.3949 |
| embryo transferred | 1 [1;1.5] | 1 [1;2] | 1.3 [1;2] | 0.1872 |
| **IMSI** <br> **cycles *n* (%)** | **17 (53.1)** | **15 (46.9)** | **0 (0.0)** | |
| retrieved oocytes | 9 [7;15] | 11 [6;14] | / | 1.0000 |
| MII oocytes | 7 [6;10] | 8 [5;12] | / | 0.8496 |
| available embryos | 3 [2;5] | 3 [2;7] | / | 0.9083 |
| embryo transferred | 2 [1;2] | 1 [0;2] | / | 0.3268 |

Values are expressed as median and quartiles [q1;q3]. *p<0.05* is considered significant.

ICSI = intracytoplasmic sperm injection; IMSI = intracytoplasmic morphologically selected sperm injection; IUI = intrauterine insemination; IVF = conventional in vitro fertilization; MII = metaphase II oocyte; PIT = pre-IMSI test

**Table 3. Main clinical outcomes depending on the ART technique used in the three PIT score subgroups.**

| | PIT score I | PIT score II | PIT score III | *p* | Global % rate in our Center for the same period |
|---|---|---|---|---|---|
| **IUI** | | | | | |
| number of transfers | *n = 32* | *n = 82* | *n = 203* | | *n = 512* |
| ongoing pregnancy *n (%)* | 2 (6.3%) | 6 (7.3%) | 10 (4.9%) | 0.7673 | 17% |
| miscarriage *n (%)* | 2 (100%) | 4 (66.7%) | 3 (30%) | 0.2053 | 23.5% |
| live birth *n (%)* | 0 (0%) | 2 (2.4%) | 7 (3.5%) | NC | 12.8% |
| **IVF** | | | | | |
| number of cycles | *n = 7* | *n = 23* | *n = 99* | | *n = 417* |
| number of transfers | *n = 9* | *n = 45* | *n = 164* | | *n = 765* |
| ongoing pregnancy *n (%)* | 2 (22.2%) | 6 (13.3%) | 40 (24.4%) | 0.3838 | 33.4% |
| miscarriage *n (%)* | 2 (100%) | 5 (83.3%) | 16 (40%) | 0.0348 | 28.1% |
| live birth *n (%)* | 0 (0%) | 1 (2.2%) | 24 (14.6%) | NC | 23.4% |
| **ICSI** | | | | | |
| number of cycles | *n = 32* | *n = 52* | *n = 95* | | *n = 753* |
| number of transfers | *n = 61* | *n = 95* | *n = 207* | | *n = 1379* |
| ongoing pregnancy *n (%)* | 9 (14.8%) | 37 (39%) | 55 (26.6%) | 0.0054 | 33.4% |
| miscarriage *n (%)* | 2 (22.2%) | 16 (43.2%) | 30 (54.4%) | 0.1872 | 27.5% |
| live birth *n (%)* | 7 (11.5%) | 21 (22.1%) | 25 (12.1%) | 0.0614 | 23.2% |
| **IMSI** | | | | | |
| number of cycles | *n = 17* | *n = 15* | *n = 0* | | *n = 48* |
| number of transfers | *n = 17* | *n = 11* | *n = 0* | | *n = 55* |
| ongoing pregnancy *n (%)* | 5 (29.4%) | 2 (18.2%) | - | 0.571 | 22.9% |
| miscarriage *n (%)* | 1 (20%) | 0 (0%) | - | 1.0000 | 12.7% |
| live birth *n (%)* | 4 (23.5%) | 2 (18.2%) | - | 0.7677 | 17.1% |

Values for ongoing pregnancies and live births are in percentages <u>per transfer</u>. Values for miscarriages are in percentages <u>per ongoing pregnancy</u>.

ART = assisted reproductive technology; ICSI = intracytoplasmic sperm injection; IMSI = intracytoplasmic morphologically selected sperm injection; IUI = intrauterine insemination; IVF = conventional in vitro fertilization; NC = statistical non convergence; PIT = pre-IMSI test

(p = 0.0614). Miscarriage rates of IVF attempts were statistically different according to PIT score (p = 0.0348). In order to better interpret our study results, the last column of Table 3 shows the global percentage of all attempts at our ART Center during the study period (Feb

**Table 4. Summary table comparing the different OR of clinical pregnancies according to ART technique between the PIT subgroups.**

| | PIT score I | | PIT score II* | | PIT score III* | |
|---|---|---|---|---|---|---|
| **ART techniques** | **OR** | **CI95%** | **OR** | **CI95%** | **OR** | **CI95%** |
| *ICSI vs. IVF* | 0.575[a] | [0.091; 3.635] | 4.651[b] | [1.687; 12.821] | 1.186[a] | [0.713; 1.974] |
| *IMSI vs. IVF* | 1.494 | [0.202; 11.046] | 1.729 | [0.279; 10.697] | - | - |
| *IVF vs. IUI* | 4.893 | [0.533; 44.903] | 2.234 | [0.642; 7.769] | 6.611 | [3.113; 14.04] |
| *ICSI vs. IMSI* | 0.385[a] | [0.104; 1.428] | 2.69[b] | [0.526; 13.748] | - | - |
| *ICSI vs. IUI* | 2.815 | [0.544; 14.557] | 10.387 | [3.966; 27.208] | 7.844 | [3.776; 16.294] |
| *IMSI vs. IUI* | 7.313 | [1.182; 45.231] | 3.861 | [0.643; 23.204] | - | - |

ART = assisted reproductive technology; CI95% = 95% confidence interval for OR; ICSI = intracytoplasmic sperm injection; IMSI = intracytoplasmic morphologically selected sperm injection; IUI = intrauterine insemination; IVF = conventional in vitro fertilization; OR = odds ratio; PIT = pre-IMSI test

* = rates of ongoing pregnancies were statistically different overall between interventions in PIT score subgroups 2 and 3 (p<0.0001)

[a,b] = OR on the same line with different letters are significantly different at the p<0.05 threshold

2017—Oct 2020). The only exclusion criterion of this reference population was women over 39 years of age.

Table 4 completes the results of Table 3 by adjusting clinical pregnancy results of PIT score subgroups on ART techniques and adding (1) two-by-two ART techniques comparisons in each PIT score subgroup estimating ORs and (2) comparisons of these different ORs of clinical pregnancy rates between the different PIT score subgroups. Results of the ICSI *vs.* IMSI comparison in PIT score I and II subgroups were significantly different and opposed. We observed that in PIT score II subgroup, clinical pregnancies rate appears to be higher with ICSI than with IMSI (OR = 2.69, CI95% = [0,53; 13.75]), whereas in the PIT score I subgroup there appears to be the reverse (OR = 0.385, CI95% = [0,10; 1.43]). In the PIT score II subgroup, more pregnancies occurred with ICSI than with IVF (OR = 4.65, CI95% = [1.69; 12.82]). The results of the ICSI *vs.* IVF comparison in the PIT score I (OR = 0.58, CI95% = [0.09; 3.64]) and III subgroups (OR = 1.19, CI95% = [0.71; 1.97]) were significantly different from PIT score II (OR = 4.65, CI95% = [1.69; 12.82]).

The other comparisons (IVF *vs.* IUI, ICSI *vs.* IUI, IMSI *vs.* IVF and IMSI *vs.* IUI) gave results that were statistically equivalent between the three PIT score subgroups.

## Discussion

The present study shows the relevance of our PIT classification for exploring apparently normal/subnormal spermatozoa in order to evaluate which assisted reproduction technique might be most appropriate for each couple selected. In developing this test, the laboratory staff wanted to obtain a fast, reproducible test to be used by anyone concerned.

We first validated the PIT methodology. We analyzed the PIT profiles of 402 patients with a normal or mild sperm abnormality on their basic semen examination [41]. We initially began with the classification by Vanderzwalmen *et al.* [5] which separates spermatozoa into four grades: grade 1: absence of vacuole, grade 2: maximum of two small vacuoles, grade 3: more than two small vacuoles or at least one large vacuole and grade 4: large vacuoles associated with abnormal head shapes or other anomalies. We voluntarily excluded morphology as a criterion for the PIT because normally-shaped spermatozoa can be chosen easily, even at low magnification. By excluding the analysis of morphology during the test, we only retained Vanderzwalmen's grades 1, 2 and 3 and focused on the evaluation of vacuoles and inter-operator comparisons. Despite this adjustment, the inter-operator coefficient variations (CVs) were still unacceptable. We thus reduced the number of grades by merging Vanderzwalmen's grades 1 and grade 2 to create a grade A group, considered as the normal spermatozoa group (Fig 1). Vanderzwalmen's grade 3, correspond to grade B sperm cells in this study and was allocated to the abnormal spermatozoa group (Fig 1). This classification allowed us to considerably reduce the time taken to perform the test and normalize the inter-operator CVs. On average, it takes two observers 15 minutes to complete the test.

In a second stage, we subsequently tried to establish reference thresholds to help biological and clinical decision-making. During the four years of PIT test validation (2017–2020), we implemented the test in our laboratory routine, once we had reproducible, reliable numerical results. We then tried to determine PIT reference thresholds and indications, starting from existing publications on the topic [5, 42]. Wittemer *et al.* [42] obtained better fertilization and implantation rates when the sperm had at least 8% normal spermatozoa compared to those with less than 8%. Based on this paper, we selected 248 couples who had ART attempts from the initial 402 who had a PIT test and arbitrarily established four PIT score categories according to the percentage of grade A sperm cells: ≤8%, 9–13%, 14–15%, and ≥16%. Observing the ART results of the two intermediate categories led us to merge them and finally individualize three PIT score subgroups (namely ≤8%, 9–15%, and ≥16%).

In a third stage, the possible relationship between PIT score results and ART outcomes was retrospectively examined. During this whole study period, the choice of ART technique was only made according to the usual clinico-biological criteria and not according to the PIT score. The aim of this study was to determine whether having a normal/subnormal sperm from a particular PIT score subgroup and a specific background would increase the chances of success in a given ART technique. This is why not only IVF techniques but also IUI were included in the study. Our method now has two advantages: (1) its simplification, which reduces the time-consuming nature of the test, especially on the day of oocyte retrieval, and (2) its reproducibility: each test is systematically read by at least two experienced operators in order to obtain reliable CVs. Another observation was that, even if teratozoospermia was present, the PIT score could be normal. On the other hand, when the PIT was abnormal, teratozoospermia was systematically found. This may be explained by the fact that vacuoles are one of the abnormalities to check for in the WHO classification [41]. Consequently, when semen analysis shows isolated teratozoospermia, we systematically propose a PIT test to these patients to further explore their sperm characteristics.

The interest of MSOME or IMSI has been widely studied [2, 35, 36, 47]. Curiously, only a few studies include patients with normal/subnormal sperm parameters or unexplained infertility [36]. In 2013, Marci et al. [38] described that, in an unselected infertile patient population, there was no significant difference in fertilization, implantation or pregnancy rates between IMSI and ICSI. More recently, in a retrospective study, Asali et al. [48] examined the efficacy of IMSI in male subfertility when conception did not occur after at least three cycles of IUI and one cycle of ICSI. Patients were divided into three subgroups according to the percentage of normal spermatozoa at MSOME: no normal spermatozoa, 0.5% to 1.5% normal spermatozoa, at least 2.0% normal spermatozoa. They concluded that IMSI provided an advantage and should be recommended for men with <2.0% normal spermatozoa according to their MSOME classification to improve ART outcomes. It is difficult to compare our results with those of Asali et al. [48] due to the large difference in methodology in the two high magnification sperm tests. However, in this work, we confirm the interest of using IMSI for patients with few or no sperm parameter abnormalities when the normal MSOME sperm count is below a certain threshold. The subgroup threshold differences between Asali et al. [48] and our study might be explained by the fact that sperm morphology was not taken into account in our classification.

Our work is original because it compares each ART technique outcome with PIT scores in a selected population (Table 3). Indeed, for these couples, when ICSI was the technique used, our results show that the PIT II subgroup had better pregnancy rates (p = 0.0054) and a tendency to better birth rates. When considering IUI, IVF and IMSI techniques, there appeared to be no PIT subgroup-related effect on clinical pregnancy outcomes. On the other hand, miscarriage rates after IVF were statistically different according to PIT subgroup, with 100% of miscarriages observed when the sperm was of PIT subgroup I category. In comparison, the miscarriage rate in the PIT subgroup III when using IVF was 40%, which is still high but probably related to other factors (couples eligible for PIT testing, polycystic ovarian syndrome, history of multiple spontaneous miscarriages. . .). Similarly, there was a 100% miscarriage rate after IUI in the PIT score I group, 66.7% with PIT score II and 30% with PIT score III, although these differences between the three PIT subgroups were statistically not significant potentially because of a lack of statistical power. All nuclear alterations of vacuolated spermatozoa described in the introduction can negatively influence male fertility and embryonic development, induce miscarriages (a decondensed DNA is more vulnerable), and eventually health problems in the offspring [29, 49]. Indeed, IMSI appears to significantly increase blastocyst development [5], decrease the incidence of structural defects [50] and miscarriage [51–56] even if the benefits of IMSI are still debated [57, 58].

The comparison of Table 3 results with the clinical pregnancy ORs between PIT subgroups according to the ART technique used (Table 4) support the use of IMSI rather than ICSI when the sperm is PIT score I, of ICSI rather than IMSI when the sperm is PIT score II and either ICSI or IVF when the sperm is PIT score III. In addition, clinical pregnancies were more frequent after ICSI than after IVF in the PIT II subgroup (ICSI *vs.* IVF: OR = 4.651, p<0.05). ICSI therefore seems to have an advantage of over IVF when the sperm is PIT score II but no significant advantage of ICSI or IVF when the sperm is PIT score I or III.

In fact, IVF should only be proposed when the sperm is of PIT score III quality, given the very high miscarriage rates with PIT score I (100%) and PIT score II (83.3%) and as it is easier and less time-consuming. Concerning IUI, our results showed that these patients had similar clinical pregnancy rates, whatever the sperm category, but in PIT score Group I, no live births were obtained. The IUI technique should therefore be avoided with PIT score I sperm, and not recommended with PIT score II, given the high rate of miscarriages. Analyzing the characteristics of the couples who ended up with miscarriages, the only factors that could explain these spontaneous abortions could be the increased number of women with polycystic ovarian syndrome in subgroup III and the fact that our results were obtained from a selected population with poorer prognoses (see study design). We are aware that there is a big difference between IUI and the other three assisted reproduction technique success rates. However, as the sperm quality in this study allowed any ART technique, the use of IUI remains a simple, valid first line treatment for young couples with unexplained infertility in the PIT score III category, even though this subject is often debated [50].

Numerous discrepancies exist in the literature, notably concerning IMSI effectiveness compared to ICSI, in improving ART results (fertilization / clinical pregnancy / live births rates / miscarriage). However, a recent systematic review and meta-analysis of an average of 4,000 live births with congenital anomalies after IMSI concluded that this technique appears to be an effective tool for reducing the incidence of structural anomalies compared to ICSI [59]. On the other hand, IMSI was shown not to alter the incidence of chromosomal anomalies i.e. trisomy 13, 18, 21 and triple X. Moreover, Bendayan *et al.* have recently explored an innovative approach to the use of IMSI for the management of infertile couples [28]. They demonstrated that high-magnification microscopy could become a simple tool to estimate the sperm epigenetic profile in real time. The epigenetic profile of human morphometrically normal spermatozoa varies according to the presence or absence of sperm-head vacuoles. While awaiting the development of more specific sperm biomarkers [60], using PIT scores remains an interesting, reliable option not to be neglected when caring for infertile couples.

Our study has certain limitations. Firstly, the heterogeneity of the numbers studied within the different ART techniques and PIT groups means that some analyses were carried out on small sample sizes, which do not always guarantee sufficient statistical power to highlight a difference of interest. Our results should therefore be taken with caution and confirmed by larger studies. Similarly, ART techniques were only compared within the PIT groups on the criterion of clinical pregnancy, and not on the criterion of live birth, which is the real aim for infertile couples. In our study, the number of live births was insufficient to allow analysis, and it is difficult to state with certainty that what is observed for clinical pregnancies will also be observed for live births. Here again, our results need to be confirmed by larger studies.

## Conclusions

In this study we were able to validate a simplified pre-IMSI test, with international quality standards, in line with the scientific literature on the subject. The number of transfers in each ART and PIT score groups remains low and obliges us to remain cautious in interpreting these

preliminary results. However, the results of this study appear to support the value of PIT for defining the choice of the most effective assisted reproduction technique in couples with normal/subnormal sperm. IMSI appears to be the technique of choice when sperm belongs to subgroup I, ICSI when it belongs to subgroup II and IVF when sperm is of subgroup III quality. If IUI is indicated, it is preferable to perform this technique mainly on subgroup III sperm. The initial reference thresholds we established have now become "decision thresholds" in our laboratory to avoid unnecessary IMSI for these couples. Today, we routinely use PIT in our ART laboratory for the following indications: isolated teratozoospermia, long periods of unexplained infertility, patients with failed IUI, two or more IVF attempts at other IVF Centers, fragmented embryos or lack of blastulation. The earlier PIT is performed, the more beneficial it is for patients. Ideally, PIT could be routinely performed as soon as the sperm is normal/subnormal or in cases of isolated teratozoospermia, before starting any first-line treatment. We believe that this strategy, which is routinely used in our laboratory, could avoid unnecessary treatment and shorten the time to delivery. These preliminary results need to be confirmed later by a prospective study (article in progress).

## Supporting information

**S1 Data.**
(XLSX)

## Acknowledgments

The authors wish to thank Teresa Sawyers, Medical Writer and Christophe Demattei Methodologist/Biostatistician at the B.E.S.P.I.M. (Department of Biostatistics, Epidemiology, Public Health and Innovation in Methodology), Nîmes University Hospital, France, for their valuable help with this article. The authors are also grateful to Nathalie Aurran, Maylis Duchassin and Pascale Buzançais, technicians at our ART laboratory in the Department of Reproductive Medicine, Nîmes University Hospital, France.

## Author Contributions

**Conceptualization:** Nathalie Rougier.

**Data curation:** Julien Sigala, Nathalie Rougier.

**Formal analysis:** Julien Robert, Thibault Mura.

**Funding acquisition:** Julien Sigala, Nathalie Rougier.

**Investigation:** Nathalie Rougier.

**Methodology:** Nathalie Rougier.

**Project administration:** Nathalie Rougier.

**Resources:** Nathalie Rougier.

**Software:** Julien Sigala, Nathalie Rougier.

**Supervision:** Nathalie Rougier.

**Validation:** Julien Sigala, Nathalie Rougier.

**Visualization:** Julien Sigala.

**Writing – original draft:** Julien Sigala.

**Writing – review & editing:** Julien Sigala, Sophie Poirey, Julien Robert, Olivier Pouget, Thibault Mura, Stephanie Huberlant, Nathalie Rougier.

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
