## [Decision Letter · Decision Letter 0]

23 Apr 2024

PONE-D-24-02211First-line infertility treatment in normal or subnormal sperm: interest of a simplified pre-IMSI testPLOS ONE

Dear Dr. SIGALA,

Thank you for submitting your manuscript to PLOS ONE. After careful consideration, we feel that it has merit but does not fully meet PLOS ONE’s publication criteria as it currently stands. Therefore, we invite you to submit a revised version of the manuscript that addresses the points raised during the review process.

Please submit your revised manuscript by Jun 07 2024 11:59PM. Please include the following items when submitting your revised manuscript:A rebuttal letter that responds to each point raised by the academic editor and reviewer(s). You should upload this letter as a separate file labeled 'Response to Reviewers'.A marked-up copy of your manuscript that highlights changes made to the original version. You should upload this as a separate file labeled 'Revised Manuscript with Track Changes'.An unmarked version of your revised paper without tracked changes. You should upload this as a separate file labeled 'Manuscript'.

We look forward to receiving your revised manuscript.

Kind regards,

Joël R Drevet, Ph.D.

Academic Editor

PLOS ONE

Journal Requirements:

Additional Editor Comments:

Two reviewers with considerable expertise in the field have evaluated this work and found that it could be improved.

They have provided several lines which, if taken into account, will strengthen the study and its conclusions.

It is recommended that the authors pay particular attention to the various remarks (especially those of reviewer 1) and modify their manuscript accordingly.

Reviewers' comments:

Reviewer's Responses to Questions

**Comments to the Author**

1. Is the manuscript technically sound, and do the data support the conclusions?

Reviewer #1: Partly

Reviewer #2: Yes

2. Has the statistical analysis been performed appropriately and rigorously? 

Reviewer #1: Yes

Reviewer #2: Yes

3. Have the authors made all data underlying the findings in their manuscript fully available?

Reviewer #1: Yes

Reviewer #2: Yes

4. Is the manuscript presented in an intelligible fashion and written in standard English?

Reviewer #1: Yes

Reviewer #2: Yes

5. Review Comments to the Author

Reviewer #1: The paper from Sigala et al. aims to develop a faster, simplifie MSOME technique for selected infertile males with apparently normal/subnormal sperm and in their history repeated ART failure (IUI, IVF), miscarriages and abnormal embryonic development.

They only assessed the presence of vacuoles in sperm heads and split patients into three groups according to the percentage of normal spermatozoa considering vacuole size and number.

They concluded that IMSI appears to be recommended when sperm belongs to the lower PIT score, ICSI to the intermediate PIT score and IUI to the higher PIT score.

Even if this study tries to identify predictive factors of sperm quality based on MSOME analysis, this retrospective study has several limitations :

- the number of infertile couples included in the study and thereafter in the different groups of ART and secondly of PIT score. This parameter is insufficiently discussed and taken into account. The selected population for PIT analysis is very heterogeneous form repeated ART failures , repeated miscarriages, abnormal embryonic development.

The methodology for MSOME analysis is insufficiently described. The authors did not describe how they differentiate large vacuoles from small ones. These different parameters should be addressed.

It should be noticed that it is difficult for the reader to find what is a subnormal semen sample. This parameter should be addressed

The analysis of semen parameters according to WHO should be more precisely described

Considering ART, IVF and ICSI cycles, no information is available concerning the ovarian stimulation, number of FSH units, estradiol at trigger, number of top quality embryos, the classification used for embryo grading for IVF, ICSI and IMSI

the females confounding factors during ovarian stimulation are not taken into account

When considering the main clinical outcomes, the authors reported the global percentage rate at their center. However, they did not describe their reference population (sample size, females and males characteristics, number of ART cycles, ...)

In Table 3, they also presented the number of transfers but they did not present for IVF, ICSI and IMSI the number of cycles of ovarian puncture. Consequently, the presentation of the data of the ART cycles did not respect the conventional presentation generally proposed in most of papers.

the reviewer was very surprised by the miscarriage rate who was very higher in the different Pit score and IUI, ICSI and IVF

The conclusion of the advantage of one technique from another based on the PIT score should be discussed carefully because the number of transfer in each ART and PIT score groups remains very low.

Major revisions are required mainly in the materials and methods and results parts. The discussion part should have a part on strength and limitation of the study

The conclusion should be more nuanced

Reviewer #2: This is a robust study with a good patient cohort size. The quite unusual approach to focus on normal/subnormal samples is warranted as there is currently no good method to triage these patients into the most appropriate assisted reproduction technique. This has led many clinics to default to using ICSI where there may not be a need for this. The use of the pre-IMSI test (PIT) may provide this unmet need, especially in recommending either IMSI or ICSI.

Although it would be more compelling to see a significant impact on live birth rate, the n number for the PIT score 1 category was understandably lower and the results are still very robust. Data were collected in a highly controlled manner and analysed appropriately. The authors should be commended on how they have controlled for female factor (as best possible), BMI and age. This is difficult to achieve and still attain a high patient number for the study in the clinical setting.

I have very minor suggestions to improve the manuscript but overall I think this forms a good additional practise to consider for clinics to triage patients and warrants publication to communicate the best use and optimisation strategy for this practise.

Suggested changes:

Could the authors please include some further representative images in Figure 1 for Grade A and B? Specifically could you show the range of severity for Grade B? Also it would be useful to highlight some cells that would fit the category ‘teratozoospermic but with normal PIT score’, this was slightly unclear. It would be more useful to have this figure ahead of the first results table in the results text or in the methods, rather than later in the discussion since you already discuss the grades in the methods section.

The potential relationship between sperm epigenetic profiles and vacuoles will be new to many readers but the authors give this almost as an afterthought to the study in the last paragraph of the discussion. I think a further expansion of the literature around vacuoles and why they might help to predict success of assisted reproduction is warranted in the introduction. Especially given the controversy of sperm vacuoles in the field, it would be good to provide further explanation up front.

Rating of the PIT scores – Just an observation but It was not intuitive to me to have PIT score I as poorest quality and PIT score III as highest quality. Perhaps something to consider ahead of publication.

In general, we face in the field that many interventions in the clinic may help to improve or predict pregnancy rate but do not necessarily translate to live birth outcomes. Although your study is still very meaningful with approaching significance, could you please comment on why this may be the case for your specific study and how your approach could be further improved? This can be included in the discussion.

The authors state that a part of the purpose of the study is to increase efficiency/decrease time take for scoring. Perhaps it is useful then to indicate how long it takes to score a sample on average? If I was reading this and considering taking up the technique I would like to know how long it takes once well optimised.

Most of the first paragraph and also second paragraph of the discussion is a repetition of the methods section, it should be revised.

Could you please comment on the relationship between sperm vacuoles and miscarriage. Is this also potentially related to epigenetic status of the sperm?

Line 424 – Reversely can be replaced with conversely

Line 451 vs line 434 – can you please double check the consistency of this recommendation. In line 434 you suggest PIT score III would recommend IVF or ICSI but then in line 451 you suggest IVF or IUI. I think based on the IUI results you present in the tables it would be difficult to recommend this but either way these statements need to be consistent for your recommendation to be meaningful.

It is good practise that you have included the data in Excel for publication. Please consider the utility of your column headings. It would help to provide a key in a second excel sheet/tab for certain terms that might need explanation.

6. PLOS authors have the option to publish the peer review history of their article (what does this mean?). If published, this will include your full peer review and any attached files.

Reviewer #1: No

Reviewer #2: **Yes: **Elizabeth Bromfield

---

## [Author Response · Author response to Decision Letter 0]

6 Jun 2024

Dears reviewers and editor,

Thanks to all of you for your comments, which we believe have helped to enrich and mature this article.

Best regards

---

## [Decision Letter · Decision Letter 1]

1 Jul 2024

First-line infertility treatment in normal or subnormal sperm: interest of a simplified pre-IMSI test

PONE-D-24-02211R1

Dear Dr. Sigala,

We’re pleased to inform you that your manuscript has been judged scientifically suitable for publication and will be formally accepted for publication once it meets all outstanding technical requirements.

Kind regards,

Joël R Drevet, Ph.D.

Academic Editor

PLOS ONE

Reviewers' comments:

Reviewer's Responses to Questions

**Comments to the Author**

1. If the authors have adequately addressed your comments raised in a previous round of review and you feel that this manuscript is now acceptable for publication, you may indicate that here to bypass the “Comments to the Author” section, enter your conflict of interest statement in the “Confidential to Editor” section, and submit your "Accept" recommendation.

Reviewer #2: All comments have been addressed

2. Is the manuscript technically sound, and do the data support the conclusions?

Reviewer #2: Yes

3. Has the statistical analysis been performed appropriately and rigorously? 

Reviewer #2: Yes

4. Have the authors made all data underlying the findings in their manuscript fully available?

Reviewer #2: Yes

5. Is the manuscript presented in an intelligible fashion and written in standard English?

Reviewer #2: Yes

6. Review Comments to the Author

Reviewer #2: (No Response)

7. PLOS authors have the option to publish the peer review history of their article (what does this mean?). If published, this will include your full peer review and any attached files.

Reviewer #2: **Yes: **Elizabeth Bromfield

---

## [Editor Report · Acceptance letter]

4 Jul 2024

PONE-D-24-02211R1 

PLOS ONE

Dear Dr. Sigala, 

I'm pleased to inform you that your manuscript has been deemed suitable for publication in PLOS ONE. Congratulations! Your manuscript is now being handed over to our production team.

Kind regards, 

on behalf of

Prof. Joël R Drevet 

Academic Editor

PLOS ONE